energy/chemical engineering

porous media, diffusion combustion, filtration combustion, packed bed length

**Author for correspondence:**
Mingming Mao
e-mail: maomm2019@163.com

# Experimental and numerical studies on combustion characteristics of N₂-diluted CH₄ and O₂ diffusion combustion in a packed bed

## Houping Li, Junrui Shi, Mingming Mao and Yongqi Liu

School of Transportation and Vehicle Engineering, Shandong University of Technology, Zibo, Shandong 255049, People's Republic of China

HL, 0000-0003-3827-6662; MM, 0000-0003-2372-8451

Experimental and numerical studies were conducted to determine the combustion characteristics of gas diffusion combustion in a porous combustor packed with 2.5 mm or 3.5 mm alumina pellets, special attention being focused on the effect of packed bed height ($h$) on combustion, NO and CO emissions. The pollutant emission of diffusion filtration combustion is studied with different packed bed lengths in the range of 40 mm $\leq h \leq$ 240 mm, fixed excess air ratio of 1.88 and fixed gas inlet velocity of 0.06 m s$^{-1}$. Results show that both immersed and surface flames coexist in the combustor. Although porous media enhance the mixing and diffusion processes, the diffusion flame shape is still observed from the side and top views of the combustor, and the diffusion filtration retains properties of diffusion combustion. The immersed flame is always observed with increase in $h$, whereas the height of surface flame decreases. The NO emission decreases sharply when $h$ is increased from 40 mm to 120 mm. However, the NO emission decreases slightly when $h > 120$ mm. In the investigated range of $h$, it is shown that $h$ has a significant influence on the CO emission, an increase in $h$ leading to a constant increase in CO for the combustors packed with 2.5 mm or 3.5 mm pellets. The maximum CO emission is 662 ppm and the minimum value is 67 ppm. In the scope of this study, the temperature on the external wall of the combustor reaches 434–513°C.

## 1. Introduction

As a new type of combustion technology, porous medium combustion has many advantages, such as high combustion efficiency, low pollutant emission and wide flammability range.

With the decrease of traditional energy sources, porous media combustion technology plays an increasingly important role in saving energy and protecting the environment. Among them, great progress has been made in the research of premixed filtration combustion [1–7]. However, there are few reports on diffusion filtration combustion, and basic research work is still needed.

The research of diffusion filtration combustion mainly focused on radiation thermal efficiency, flame shape, combustion stability and pollutant emission. Early studies on the thermal efficiency focused on the liquid fuel porous medium burners [8,9]. Then the radiation thermal efficiency of gas-fuelled porous medium burners has become the focus of research [10,11]. Kamal *et al.* [10] pointed out that the relative position of flame front in a porous medium burner has a significant influence on combustion. They experimentally studied the radiation thermal efficiency of the burner with and without porous media. The results showed that the heat radiation flux was 5.7 times by adjusting the distance between fuel nozzle and porous media and swirl intensity. Dobrego *et al.* [11] modelled $CH_4$/air combustion in a pellet packed bed using a two-dimensional two-temperature model. It was found that stable combustion can also be achieved at a fairly low equivalent ratio, but the simulation results were not compared with the experimental results.

The flame shape of diffusion filtration combustion has always been the focus of researchers' attention. Kamiuto *et al.* [12,13] experimentally studied the flame shape in circular and square burners and constructed flame shapes from the burner surface. They analysed the flame structures using a flame sheet model. Due to the limitation of the experimental device, the overall flame shape cannot be observed from the combustor side, and the combustion characteristics at different height of the packed bed have not been revealed in the experiment.

Dobrego *et al.* [11] confirmed by numerical simulation that, unlike the premixed filtration combustion, the temperature difference between gas and solid in the combustion zone is very large. Shi *et al.* [14] studied the diffusion combustion characteristics in a pellet packed bed using a two-dimensional two-temperature model. The results showed that the high-temperature zone of gas and solid phases exists in different positions of burner, but the thermal non-equilibrium between gas and solid phases outside the high-temperature zone was very small. Studies by Shi *et al.* [14,15] and others show that the flame height was mainly controlled by the fuel diffusion rate.

The results showed that the diffusion combustion in porous media exhibits high stability [16–19]. Zhang *et al.* [16] developed a 125 kW reciprocating flow porous medium diffusion combustion furnace. The results showed that, compared with the conventional gas burner without regenerator, the fuel/air can be burned stably with fuel saving of 35%. Wang *et al.* [17] experimentally studied the diffusion combustion characteristics of liquid petroleum gas/air in a foam ceramic packed bed. They found that the effect of gas flow rate on maximum combustion temperature was small, while the equivalence ratio had a significant effect, but they did not report the emission of pollutants. Kokubun *et al.* [18] used the assumption of large activation energy to theoretically analyse the combustion stability and extinction characteristics of diffusion filtration combustion, focusing on the effects of porosity and flow rate on extinction. The results showed that when the porosity is small enough, the convective heat transfer between gas and solid phases is enhanced, and a considerable part of the combustion heat is absorbed by solid phase in the combustion zone, resulting in a decrease in the temperature of the combustion zone, so that part of the fuel penetrates the flame zone without chemical reaction, and finally leads to flameout. Ning *et al.* [19] experimentally studied the diffusion combustion stability for Y-type combustors filled with and without fibre porous medium. It was found that the combustion stability was improved and the combustible range was widened when the fibre porous medium was filled in the Y-type burner. Shi *et al.* [20] studied experimentally the diffusion combustion of methane/air in pelleted bed and found that the combustion is stable.

There have been few reports on pollutant emissions from diffusion filtration combustion, most of which focused on the study of pollutant emissions in liquid fuel combustion [10]. Kaplan *et al.* [21] used heptane/air as fuel. It was found that CO emission was 3–7 ppm and $NO_x$ emission was 15–20 ppm (corrected oxygen content in flue gas was 3%). Kamal *et al.* [10] reported that not only the radiation thermal efficiency but also the ultra-low emission of $NO_x$ and CO were obtained by enhancing the mixing of fuel and oxidant. At high swirl number, CO and unburned hydrocarbons decreased significantly, whereas $NO_x$ emission was less than 10 ppm.

From the above analysis, it can be seen that there has been little research on diffusion combustion characteristics and pollutant emissions at different $h$. To develop radiation heat exchangers, it is necessary to optimize $h$ and the size of packed bed pellets. At the same time, the temperature distributions on the external surfaces of the combustor have not been revealed, which is very important to develop radiation heat exchangers by diffusion filtration combustion technology. This information is also very important for further understanding of diffusion combustion characteristics and controlling pollutant emissions.

**Figure 1.** Schematic diagram of experimental set-up. (*a*) Schematic of the combustion system and (*b*) combustor.

In this paper, an experimental set-up for study diffusion filtration is built to study the diffusion combustion characteristics of nitrogen-diluted methane and oxygen combustion in porous media. By changing $h$ and the diameter of porous media pellets, the flame shape and pollutant emissions of diffusion filtration combustion are analysed. The two-dimensional two-temperature model is used to simulate the flow, heat transfer and combustion in porous media, which provides a basis for the analysis of pollutant emissions and the development of radiation heat exchangers.

## 2. Combustion experimental system

The experimental system for diffusion combustion of nitrogen-diluted methane and oxygen in porous media is shown in figure 1. The gases are controlled and measured by a mass flow controller (MFC). Methane and nitrogen flow through the MFC and enter the mixer. Then the mixture enters the pellet packed bed through the fuel nozzle and combusts with oxygen entering the combustor through the oxygen nozzle. The diffusion chamber is made of stainless steel with high-temperature resistance. The length of the diffusion chamber is 120 mm, the width of the central fuel groove is 28 mm, the width of the oxidizer groove on both sides is 9 mm, and the thickness of the middle two baffles is 1.4 mm. In order to obtain a uniform air flow and support the top packed bed, the diffusion chamber is filled with alumina pellets with diameter of 2 mm, which can prevent flashback at the same time. The combustor is a transparent quartz glass tube with length of 490 mm, an internal diameter of 48.4 mm and thickness of 3 mm, which is filled with alumina pellets with a diameter of 2.5 mm or 3.5 mm during the experiment, and the porosity of the packed bed is 0.39. It should be pointed out that in order to observe the flame, the combustor is not insulated, which leads to a large heat loss through the tube walls, and then affects the combustion and pollutant emissions. Before the start of the experiment, the height of the packed bed in the combustor is 40 mm. When the fuel and oxidizer are fed into the combustor, the ignition is started at the combustor outlet by a torch. The positions of the cameras are fixed throughout the experiments. After the flame is stabilized, photographs of the flame are taken from the side and top of the combustor by cameras. In order to avoid destroying the porous medium structure, infrared thermal imagers (FLUKE Ti32) are used to measure the temperature of the combustor's external walls, which is used to qualitatively describe the position of the flame. Flue gas analyser (testo 350-XL) is used to analyse pollutant emissions. Then, alumina pellets are added to the combustor to ensure that $h$ is increased by 40 mm each time, this process being repeated until $h$ reaches the maximum value of 200 mm.

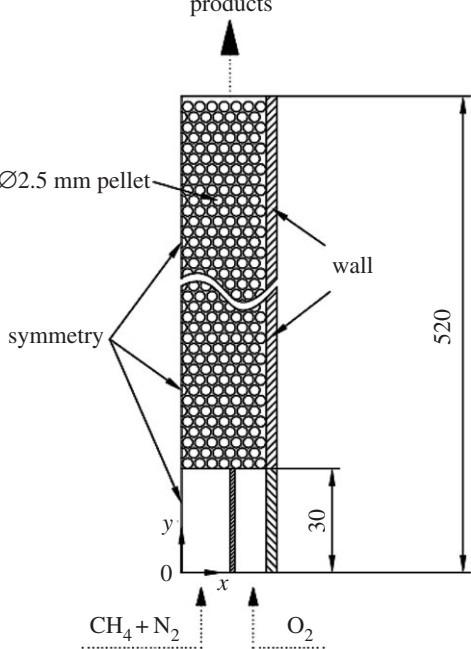

**Figure 2.** Schematic diagram of the computational domain.

# 3. Numerical simulation

Diffusion filtration combustion is investigated numerically using a two-dimensional two-temperature model based on the volume-averaged method. We use our experiment in §2 as physical model. The combustor is simplified to a two-dimensional model due to its symmetrical structure. In order to simplify the calculation, the computational domain does not include the diffusion chamber, but extends 30 mm free space upstream of the combustor inlet to simulate the flow distribution at the combustor inlet, as shown in figure 2. To allow consideration of the thermal conductivity and radiation of the quartz glass tube, the calculation domain included the quartz glass tube, giving a total calculation domain width and height of 27.4 and 520 mm, respectively.

To simplify the problem, the following assumptions are made:

(1) Inert homogeneous and optically thick porous media are assumed. The solid radiation from the solid phase is taken into account by the Rosseland approximation model [22].
(2) The gas flow in the packed bed is assumed to be laminar and gas radiation is computed by discrete model [22].
(3) The porosity in the packed bed is uniform and set to be 0.39.
(4) The diffusion of chemical species and heat in the vertical and horizontal directions is assumed to be uniform.

Based on the above assumptions, a set of differential equations can be obtained [14], as presented in the following.

*Continuity equation*

$$\nabla \cdot (\rho_g \mathbf{v}) = 0, \tag{3.1}$$

where $\rho_g$ is the gas density; $\mathbf{v}$ is the velocity vector.

*Vertical momentum equation*

$$\nabla(\rho_g \mathbf{v} u) = \nabla(\mu \nabla u) - \frac{\Delta p}{\Delta x} + \rho_g g, \tag{3.2}$$

where $u$ and $x$ are the vertical velocity and the coordinate, respectively; $p$ is the pressure; $\mu$ is the dynamic viscosity; $g$ is gravitational acceleration.

*Horizontal momentum equation*

$$\nabla(\rho_g \mathbf{v} v) = \nabla(\mu \nabla v), \tag{3.3}$$

where $v$ and $y$ are the horizontal velocity and the coordinate, respectively.

The pressure loss in vertical direction in the packed bed is computed by

$$\frac{\Delta p}{\Delta l} = 150 \frac{(1-\varepsilon)^2}{\varepsilon^3} \frac{\mu u}{d^2} + 1.75 \frac{1-\varepsilon}{\varepsilon^3} \frac{\rho_g u^2}{d}, \tag{3.4}$$

where $\varepsilon$ is the porosity; $d$ is the pellet diameter.

*Gas-phase energy equation*

$$\nabla \cdot (\rho_g c_g \mathbf{v} T_g) = \nabla \cdot ((\lambda_g + \rho_g c_g D^d)\nabla T_g) + h_v(T_s - T_g) + \sum_i \omega_i h_i W_i + q_R, \tag{3.5}$$

where $D^d$ is the thermal dispersion coefficient and is expressed as $D^d = 0.5\varepsilon d v c_g$ [1]; $T_g$, $c_g$, $\lambda_g$ are the gas temperature, the specific heat, and the thermal conductivity, respectively; $T_s$ is the solid temperature; $\omega_i$, $W_i$ are the reaction rate and the molecular weight of species $i$. The chemical mechanism for methane combustion provided by Fluent 15.0 is used in this work. To save computation cost, single-step chemistry rather than detailed chemical mechanism is applied, thus simulation of CO and $NO_x$ formation is not considered in this study. The reaction rate is computed as $\omega_{CH_4} = A(Y_{CH_4}/W_{CH_4})^{0.2}(Y_{O_2}/W_{O_2})^{1.3}\exp(-E/RT_g)$, where $A$ is pre-exponential factor with consistent units and its value is $2.119 \times 10^{11}$, $E$ is activation energy ($2.027 \times 10^8$ J kg$^{-1}$ mol$^{-1}$), and $R$ is the universal gas constant.

The convective heat transfer coefficient between the gas and solid phases in packed bed [1], $h_v$, is computed as

$$h_v = \frac{Nu_v \times \lambda_g}{d^2}, \, Nu_v = 2 + 1.1\,Pr^{1/3}\,Re^{0.6}. \tag{3.6}$$

$q_R$ is the source term due to thermal radiation.

*Solid-phase energy equation*

$$\nabla \cdot (\lambda_{eff}\nabla T_s) + h_v(T_g - T_s) = 0, \tag{3.7}$$

where $\lambda_{eff}$ is the effective thermal conductivity of the packed bed and can be expressed as $\lambda_{eff} = (1-\varepsilon)\lambda_S + \lambda_{rad}$, where $\lambda_S$ and $\lambda_{rad}$ are the solid thermal conductivity and the radiative heat transfer coefficient of the alumina pellets, respectively. Using the Rosseland approximation, $\lambda_{rad}$ is expressed as [1]

$$\lambda_{rad} = \left(\frac{32\varepsilon\sigma d}{9(1-\varepsilon)}\right)T_s^3. \tag{3.8}$$

*Species conservation equation*

$$\nabla \cdot (\rho_g \mathbf{v} Y_i) - \nabla \cdot (\rho_g (D_i + D_p)\nabla Y_i) - \omega_i W_i = 0, \tag{3.9}$$

where $D_i$ is the molecular diffusion coefficient and can be expressed as [13]

$$D_i = 0.18\left(\frac{T_g}{T_0}\right)^{1.75}\left(\frac{P_0}{p}\right), \tag{3.10}$$

where $P_0$ and $T_0$ are the ambient pressure and temperature, respectively. $D_p$ is the mass dispersion coefficient and is expressed as [13]

$$D_p = 0.1161\varepsilon d v(1-\varepsilon)^{1.4609}. \tag{3.11}$$

*Energy equation for the quartz glass*

$$\nabla \cdot (\lambda_{s1}T_{s1}) + q_r = 0, \tag{3.12}$$

where $T_{s1}$ and $\lambda_{s1}$ are the temperature and thermal conductivity of the quartz glass and $q_r$ is the source term due to radiation. The symbols used in this work are presented in table 1.

## 3.1. Boundary conditions

The boundary conditions are presented in table 2. The experimental and numerical parameters used in this work are shown in table 3.

**Table 1.** Symbols used in this work.

| nomenclature | |
| --- | --- |
| $c$ specific heat, kJ kg$^{-1}$ K$^{-1}$ | $d$ diameter of spheres, mm |
| $D_i$ molecular diffusion coefficient, cm$^2$ s$^{-1}$ | $D_p$ mass dispersion coefficient, cm$^2$ s$^{-1}$ |
| $h$ height of porous media, mm | $h_i$ molar enthalpy of species $i$, kJ kg$^{-1}$ |
| $h_v$ convective heat transfer coefficient, W m$^{-3}$ K$^{-1}$ | $p$ pressure, Pa |
| $T$ temperature, K | $T_0$ ambient temperature, K |
| $u$ axial velocity, m s$^{-1}$ | $u_{g,in}$ gas mixture velocity at inlet, m s$^{-1}$ |
| $v$ radial velocity, m s$^{-1}$ | $W_i$ molecular weight of species $i$, kg mol$^{-1}$ |
| $x$ axial coordinate, m | $X$ molar fraction |
| $y$ radial coordinate, m | $Y$ mass fraction |
| **Greek symbols** | |
| $\alpha$ excess air ratio | $\lambda$ thermal conductivity, W m$^{-1}$ K$^{-1}$ |
| $\lambda_{eff}$ effective thermal conductivity, W m$^{-1}$ K$^{-1}$ | $\lambda_{rad}$ radiation conductivity, W m$^{-1}$ K$^{-1}$ |
| $\lambda_{s1}$ thermal conductivity of quartz tube, W m$^{-1}$ K$^{-1}$ | $\rho$ density, kg m$^{-3}$ |
| $\varepsilon$ porosity | $\omega_i$ reaction rate of species $i$, mol m$^{-3}$ s |
| $\sigma$ Stephan–Boltzmann constant, W m$^{-2}$ K$^{-4}$ | $\mu$ dynamic viscosity, Pa s |
| $\varepsilon_r$ solid surface emissivity | |
| subscripts | |
| g gas | s solid |

**Table 2.** Boundary conditions.

| | |
| --- | --- |
| combustor inlet | $T_g = T_s = 300$ K; $u = 0.06$ m s$^{-1}$; $v = 0$; $Y_{CH_4} = 0.188$; $Y_{N_2} = 0.812$; $Y_{O_2} = 1$ |
| combustor outlet | $\dfrac{\partial T_g}{\partial x} = \dfrac{\partial (Y_{CH_4})}{\partial x} = \dfrac{\partial (Y_{O_2})}{\partial x} = 0$ |
| solid temperature at inlet and outlet | $\lambda_{eff} \dfrac{\partial T_s}{\partial x} = -\varepsilon_r (T_{s,out}^4 - T_0^4)$ |
| symmetry conditions ($y = 27.4$ mm) | $\dfrac{\partial T_g}{\partial y} = \dfrac{\partial T_s}{\partial y} = \dfrac{\partial Y_i}{\partial y} = \dfrac{\partial u}{\partial y} = v = 0$ |

**Table 3.** Experimental and numerical parameters in this work.

| | | | |
| --- | --- | --- | --- |
| $d$ | 2.5 mm, 3.5 mm | $h$ | 40–200 mm |
| $u_{g,in}$ | 0.06 m s$^{-1}$ | $\alpha$ | 1.88 |
| $\lambda_s$ | 0.2 W m$^{-1}$ K$^{-1}$ | $\lambda_{s1}$ | 2 W m$^{-1}$ K$^{-1}$ |
| $\varepsilon$ | 0.39 | $\varepsilon_r$ | 0.43 |

## 3.2. Initial conditions and solution

The governing equations are numerically solved using computational fluid dynamics software Fluent 15.0. To allow gas and solid phases to have different temperatures, user defined scalars are used. The SIMPLE algorithm is used to handle the pressure and velocity coupling. Mesh independence of the results is verified. The residuals of $10^{-6}$ for energy equations and $10^{-3}$ for all other equations are taken as the convergence criteria.

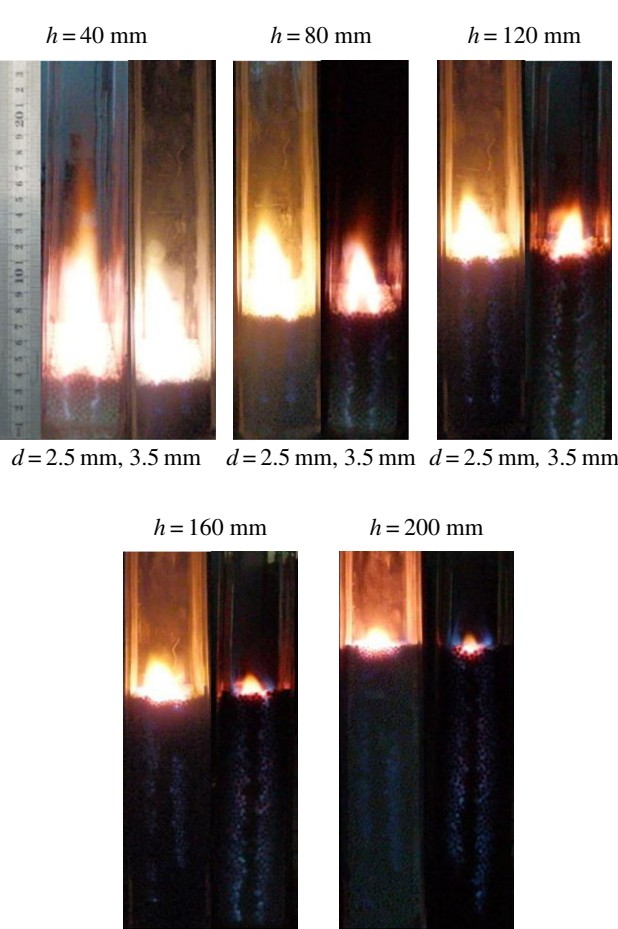

$h = 40$ mm      $h = 80$ mm      $h = 120$ mm

$d = 2.5$ mm, 3.5 mm    $d = 2.5$ mm, 3.5 mm   $d = 2.5$ mm, 3.5 mm

$h = 160$ mm      $h = 200$ mm

$d = 2.5$ mm, 3.5 mm    $d = 2.5$ mm, 3.5 mm

**Figure 3.** The diffusion flame shapes from side view of the combustor for different $d$ and $h$.

## 4. Results and discussion

### 4.1. Flame structure and temperature distribution on the outer surface of combustor

Figure 3 shows the flame shapes from side view of the combustor for different $d$ (2.5 and 3.5 mm) and $h$ (40–200 mm) at $Y_{CH4} = 0.188$ and $u_{g,in} = 0.06$ m s$^{-1}$. The total excess air ratio is 1.88. It can be seen from figure 3 that in the range of $h = 40$–200 mm, there are two kinds of flame: the flame immersed in the packed bed and the surface flame on the surface of the pellets. Two bright lights in the packed bed correspond to the chemical reaction areas. Methane enters the combustor from the fuel nozzle and mixes with oxygen from the oxygen nozzle, the combustion taking place immediately just above the combustor inlet. Two bright areas are the shape of the flame. It can be seen from the figure that the flame shape of diffusion combustion in porous medium is conical flame, similar to that of free space. There is also a conical flame on the surface of the packed bed. This may be due to the limited height of the packed bed; methane cannot be completely burned in the porous media, and continues to burn in the free space on the surface of the packed bed.

As shown in figure 3, immersed flame and surface flame always exist as $h$ increases from 40 to 200 mm. The length of immersed flame is equal to $h$, while the height of surface flame is decreasing. This is because with the increase of $h$, the residence time of methane in the packed bed increases, so the consumption of fuel in the packed bed increases. That is to say, the methane escaping to the free space decreases, so the flame height decreases accordingly. However, when $h$ increases to 200 mm, the weight of quartz glass tube reaches its limit due to the increase of $h$. For safety reasons, the maximum limit of $h$ in the experiment is 200 mm. Kamiuto *et al*. [12,13] reported the combustion characteristics of diffusion filtration in similar experimental devices, but they did not report the flame shape from the side view and pollutant emissions. At the same time, it was observed in the experiment that the height of surface flame for $d = 2.5$ mm is less than that of $d = 3.5$ mm for the same $h$. This is because

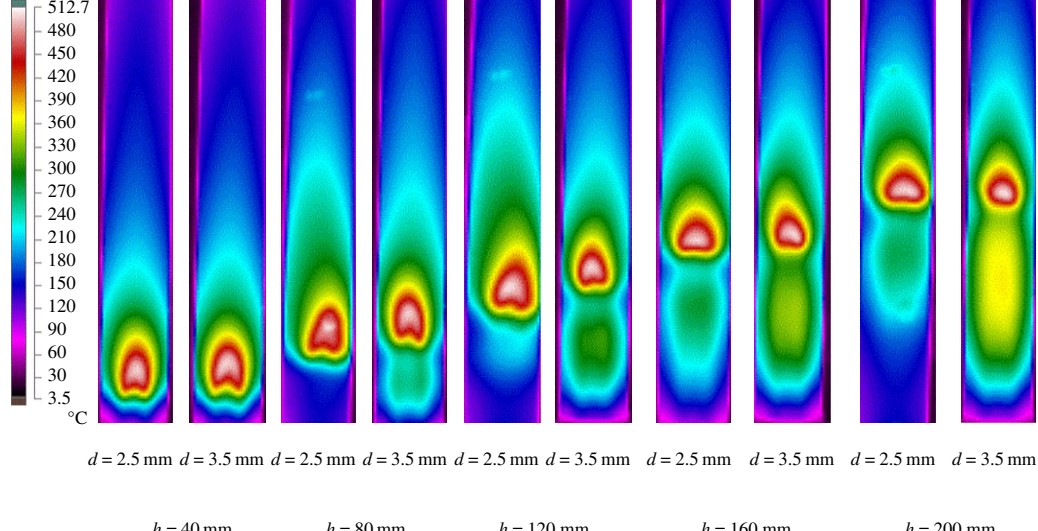

$d = 2.5\,mm$  $d = 3.5\,mm$  $d = 2.5\,mm$  $d = 3.5\,mm$  $d = 2.5\,mm$  $d = 3.5\,mm$  $d = 2.5\,mm$  $d = 3.5\,mm$  $d = 2.5\,mm$  $d = 3.5\,mm$

$h = 40\,mm$ $\qquad$ $h = 80\,mm$ $\qquad$ $h = 120\,mm$ $\qquad$ $h = 160\,mm$ $\qquad$ $h = 200\,mm$

**Figure 4.** The measured temperature profiles of external walls of the combustor for different $d$ and $h$.

with the increase of $d$, the axial dispersion coefficient increases, which enhances the transport and mixing processes for the fuel and oxidant. Therefore, the fuel consumption rate increases and the flame height decreases. According to the literature [12], the total diffusion coefficient includes molecular diffusion coefficient $D_i$ and mass dispersion coefficient $D_p$. $D_p$ is the mass dispersion coefficient and expressed as $D_p = 0.1161\varepsilon d v (1 - \varepsilon)^{1.4609}$ [17]. The above formulae include the porosity $\varepsilon$, the diameter of sphere $d$, the diffusion coefficient of gas molecules and the flow velocity $v$. The porosity of 2.5 mm and 3.5 mm spheres is about 0.39. It can be seen from the above formula that one of the most important factors affecting the total diffusion coefficient is $d$. Assuming that other variables remain unchanged, the mass dispersion coefficient increases linearly with $d$. Therefore, the radial diffusion of gas in the large diameter pellet packed bed is enhanced, which accelerates the fuel consumption. Therefore, the flame height corresponding to the 3.5 mm pellet packed bed decreases.

Figure 4 shows the measured temperature distributions of the external walls from side view of the combustor by the infrared thermal imager, which is used to qualitatively analyse the combustion in the combustor. As can be seen from figure 4, the high-temperature region propagates downstream, while the width of the high-temperature region increases with the increase of $h$ from 40 to 200 mm for $d = 2.5\,mm$ and $d = 3.5\,mm$. This is because porous media have good heat transfer properties and heat capacity compared with the gas phases. With the increase of $h$, more heat is accumulated in the packed bed. However, the maximum temperature of the external walls ($T_{wall,max}$) decreases slightly. For example, when $h$ increases from 40 to 200 mm for $d = 3.5\,mm$, $T_{wall,max}$ decreases from 469.9°C to 433.8°C. In addition, it can be seen that within the investigated experimental range, $T_{wall,max}$ corresponding to the 2.5 mm pellet packed bed is always greater than that of the corresponding 3.5 mm pellet packed bed. This may be due to the fact that the heat transfer between gas and solid phases in the packed bed with 2.5 mm pellets is stronger than that with 3.5 mm pellets, especially between the pellets and the inner wall of the combustor. This is explained by the convective heat transfer coefficient $h_v$ between the gas and solid phases: it is computed as [1] $h_v = Nu_v \times \lambda_g / d^2$, where $Nu_v$ is volumetric Nusselt number. Therefore, more heat is transferred to the side wall of the combustor, resulting in larger $T_{wall,max}$. From figure 4, it can be seen that the whole temperature distribution value is very large. In the scope of this study, the temperature on the external wall of the combustor reaches 434–513°C. It is feasible to develop a radiation heat exchanger by diffusion filtration combustion.

Figure 5 shows the flame structures from top side of the combustor by camers. As can be seen from figure 5, two bright lights, which are approximately parallel, indicate the chemical reaction areas, and their widths represent the flame width at a certain bed height from the nozzle. The distance between the two parallel lights becomes narrower with $h$. Combining with the analysis of figure 3, it can be seen that the flame structure of diffusion filtration combustion is similar to that of classical gas diffusion combustion in free space. Although the porous medium enhances the mixing and diffusion processes, the obvious chemical reaction region at the interface can still be seen.

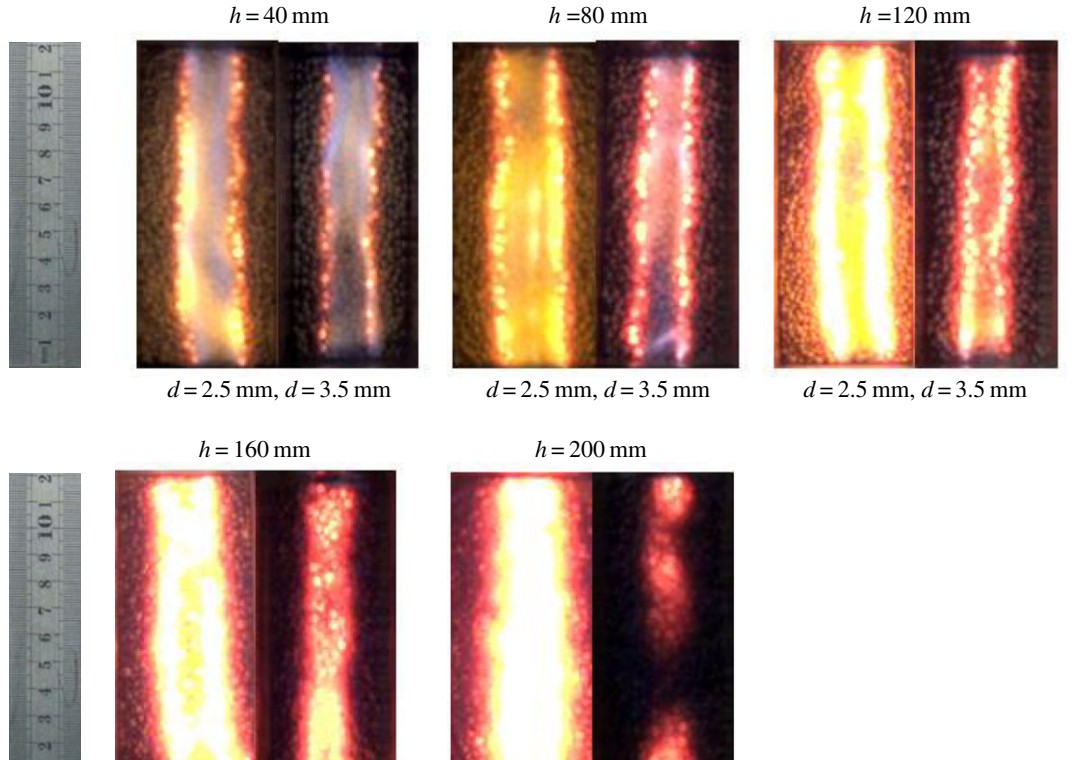

**Figure 5.** The diffusion flame shapes from upper view of the combustor for different $d$ and $h$.

## 4.2. Pollutant emissions

Exhaust gas is collected using a stainless steel probe placed at the top of the combustor, and the concentrations of $NO_x$ and CO in the exhaust from the combustor are measured. The height of the combustor and the maximum porous media length are 490 mm and 200 mm, respectively. It is observed in the experiment that the maximum flame height is about 200 mm for all investigated cases. This means that the exhaust gas is well premixed and the chemical reactions come to a halt before entering the stainless steel. Thus, the measured $NO_x$ and CO emissions are spatially averaged values.

Figure 6 shows NO, CO emissions and the maximum temperature on the external walls of the combustor ($T_{wall,max}$) for $d = 2.5$ mm, 3.5 mm and 40 mm $\leq h \leq 200$ mm. The mechanism of $NO_x$ formation for hydrocarbon fuel combustion includes thermal, fuel and prompt mechanisms. The fuel used in this experiment is methane, so there is no fuel-type $NO_x$ formation. Kamal *et al*. [10] pointed out that only when the emission of $NO_x$ is less than 9 ppm does the generation of prompt $NO_x$ become very important. When the combustion temperature is greater than 1500°C, the thermal $NO_x$ produced by combustion temperature increases rapidly. Therefore, the main mechanism of $NO_x$ formation in this study is thermal $NO_x$, which is due to the oxidation of nitrogen at high temperature. The experimental results show that the main component of $NO_x$ is NO, and other nitrogen oxides are very few. Therefore, NO is used to represent nitrogen oxides in this paper. As shown in figure 6, NO emission decreases sharply when $h$ is increased from 40 to 120 mm. Then NO emission decreases slowly as $h$ increases. This indicates that the combustion temperature decreases significantly when $h < 120$ mm. It is noted that NO emission for $d = 2.5$ mm is greater than that for $d = 3.5$ mm when $h < 120$ mm. As $h$ further increases, the effect of $d$ on NO emission vanishes. However, the influence of the height of packed bed on CO emission is opposite to that on NO emission. It is observed in figure 6$b$ that CO emission for $d = 3.5$ mm is always greater that for $d = 2.5$ mm. From figure 6$a,b$, it indicates that the combustion temperature in the combustor decreases as $h$ increases. This is partly verified by the $T_{wall,max}$ measured by the thermal imager. It is shown in figure 6$c$ that $T_{wall,max}$ always decreases when $h$ is increased from 40 to 200 mm for $d = 2.5$ mm, 3.5 mm. This may indicate that the maximum combustion temperature continually decreases as $h$ increases for $d = 2.5$ mm, 3.5 mm. Therefore, the NO emission decreases while CO emission increases as $h$ increases. The combustion temperature in the combustor will be analysed with the numerical results.

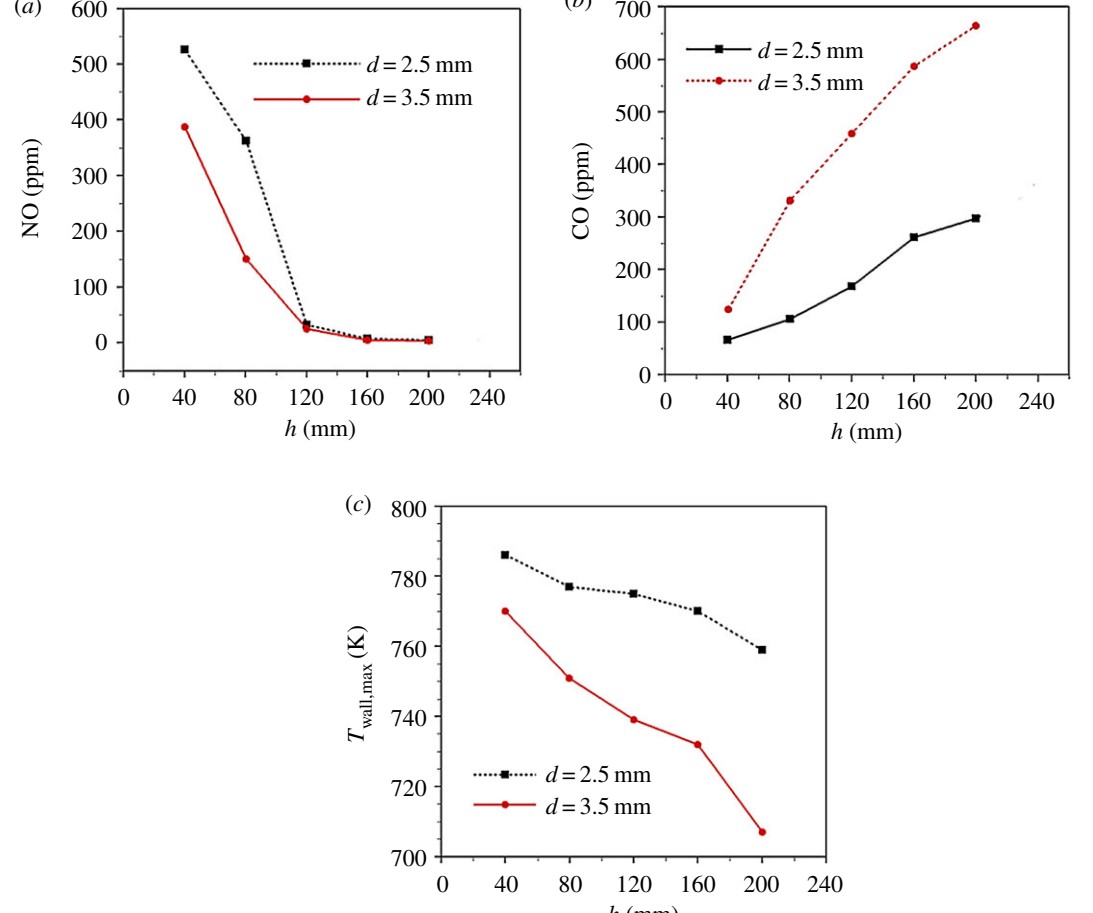

**Figure 6.** The (*a*) NO and (*b*) CO emissions and (*c*) the maximum external wall temperatures of the combustor for different *d* and *h*.

Compared with previous studies [10], the CO emissions obtained in this paper are greater; the maximum value is 662 ppm and the minimum value is 67 ppm. This may be due to the following two reasons. Firstly, for observation of flame shape the naked combustor is used in this study. This results in huge heat loss through the combustor walls to the surrounding by convection and radiation heat transfer. The radial temperature gradient of the combustor is large, and the temperature near the inner wall of the combustor is low. Therefore, the flame may be extinguished near the inner wall of the combustor, resulting in a large amount of CO. Secondly, enhanced mixing for fuel and oxygen helps to decrease CO emission, as pointed out by Kamal & Mohamad [10]. However, we do not use any enhanced gas mixing device before fuel and oxygen burns and this leads to greater CO emission.

Figure 7 shows predicted gas temperature distributions in the combustor for $d = 3.5$ mm and different $h$. The other parameters are the same as those in the previous figures. The influence of $h$ on pollutant emissions is analysed based on the predicted gas temperature distributions. It can be seen from figure 7 that one annular zone of high gas temperature is observed immediately above the fuel inlet. This indicates that chemical reaction occurs near the combustor inlet. This may be partly due to the extensive heat transfer between the gas and solid phases compared to the diffusion combustion in open space. The gas mixture is preheated by the solid phase in the packed bed, thus the high-temperature zone enlarges. Meanwhile, there is a high-temperature region in the free space above the sphere surface. With the increase of $h$, the maximum gas temperature ($T_{g,max}$) in the high temperature zone of combustion decreases rapidly. For example, $T_{g,max}$ decreases sharply from 2380 to 2047 K as $h$ increases from 40 to 120 mm. Meanwhile, the height of high-temperature combustion zone decreases with $h$. As mentioned earlier, the residence time of fuel in the packed bed increases with $h$. As a result, the fuel consumption in the porous media increases, and the methane escaping to the free space decreases accordingly. Therefore, the maximum combustion temperature in the free space decreases while the flame height decreases, which leads to a significant increase in CO emission from the combustor, while NO emission decreases rapidly.

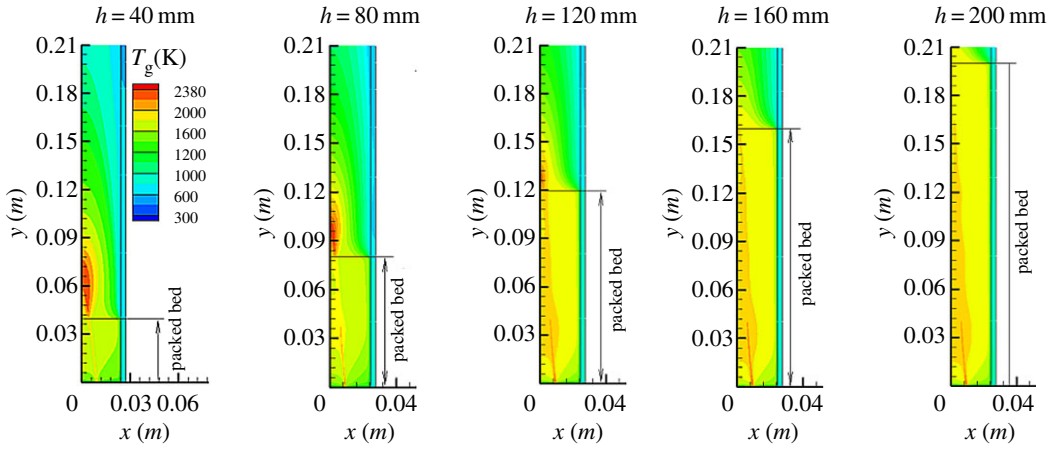

**Figure 7.** Predicted gas temperature profiles in porous combustor.

It is noted that the temperature distribution of diffusion filtration combustion is quite different from that of classical diffusion combustion in free space. It can be seen from the figure that there is a wide high gas temperature zone in the packed bed, which is due to gas combustion in porous media, and part of the reaction heat is stored in the packed bed through intense convective heat transfer between the gas and solid phases, resulting in the widening of the high-temperature zone of gas. For the classical gas diffusion combustion in free space, there exists a conical flame, which has a high temperature and a large temperature gradient on a very narrow flame surface. The diffusion combustion in porous media has a wide high-temperature region, but it can be clearly seen in figure 7 that the chemical reaction region occurs in a very narrow region.

It should be pointed out that one-step reaction mechanism instead of detailed chemical reaction mechanism is used in the numerical study, and only analyses the formation of CO and NO from the temperature distribution of combustion by numerical predictions. In addition, good thermal insulation and enhanced mixing of fuel and oxygen are necessary to reduce CO and NO emissions.

## 5. Conclusion

Experimental studies on the $N_2$-diluted methane and $O_2$ combustion in a plane-parallel porous burner were conducted with different packed bed lengths. The diffusion filtration combustion in porous media was simulated applying a two-dimensional two-temperature model with one-step chemistry. The major conclusions are as follows:

(1) In the experiment, the coexistence of immersed flame and surface flame is confirmed. With the increase of $h$, the immersed combustion always exists in the porous media, while the flame height of the surface combustion decreases.

(2) Despite the introduction of porous media into free space, obvious diffusion flame shape can be observed from the front and top sides of the combustor. Diffusion filtration combustion retains obvious diffusion combustion properties.

(3) The height of packed bed has an important influence on NO emission. With the increase of $h$, NO emission decreases rapidly when $h < 120$ mm; then the effect of $h$ on the NO emission vanishes with $h$.

(4) $h$ has an important influence on CO emission. With the increase of $h$, CO emission always increases for 40 mm $\leq h \leq$ 200 mm.

(5) The NO emission for 2.5 mm pellet packed bed is greater than that for 3.5 mm pellet packed bed when $h < 120$ mm. Then the influence of $d$ on NO emission vanishes when $h > 120$ mm.

(6) The CO emission for 3.5 mm pellet packed bed is always greater than that for 2.5 mm pellet packed bed for 40 mm $\leq h \leq$ 200 mm.

(7) The external wall temperature of the combustor reaches 434–513°C. It is feasible to develop a radiation heat exchanger by diffusion filtration combustion.

Data accessibility. Our data are deposited in the Dryad Digital Repository at: https://doi.org/10.5061/dryad.269rh68 [23].
Authors' contributions. M.M. contributed to the conception of the study. S.J. contributed significantly to analysis and manuscript preparation; L.H. performed the data analyses and wrote the manuscript; L.Y. helped perform the analysis with constructive discussions. All authors gave final approval for publication.

Competing interests. We declare we have no competing interests.

Funding. This work is supported by the National Natural Science Foundation of China (grant no. 51876107) and Shandong Natural Science Foundation Projects (grant no. ZR201709200116).

Acknowledgements. The author is grateful to many colleagues with whom he had the privilege to interact and collaborate over the years and whose work is partially referenced in this article.

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
