## [Reviewer comments · Royal Society Open Science]

Review History

RSOS-190492.R0 (Original submission)

Review form: Reviewer 1

Is the manuscript scientifically sound in its present form?

Yes

Are the interpretations and conclusions justified by the results?

Yes

Is the language acceptable?

Yes

Is it clear how to access all supporting data?

Not Applicable

Do you have any ethical concerns with this paper?

No

Have you any concerns about statistical analyses in this paper?

No

Recommendation?

Accept with minor revision (please list in comments)

Comments to the Author(s)

The manuscript reports experimental and numerical investigations on the diffusion filtration combustion in a pellet packed bed for N₂-diluted CH₄ and O₂. The findings in the paper are interesting. The study started with experimental investigation on flame structure, temperature distribution and NO_x emission. Both immersed flame and surface flame were observed in the experiment. Numerical simulation was conducted to reveal the NO_x formation mechanism with different bed lengths.

The manuscript was well organized and written with fluent English. Publication is recommended after minor revisions. My specific comments are listed below.

1. Please provide a table to include all the experimental parameters and operating conditions.
2. Some grammatical errors are found in the article, such as "It can be seen from the figure that the flame shape of diffusion combustion in porous medium is similar to that of free space, similar to that of conical flame."
3. The authors stated that "In order to consider the thermal conductivity and radiation of quartz glass tube...", however, the energy equation for the quartz glass tube was not included in the mathematics model.

Review form: Reviewer 2**Is the manuscript scientifically sound in its present form?**

Yes

Are the interpretations and conclusions justified by the results?

Yes

Is the language acceptable?

Yes

Is it clear how to access all supporting data?

Not Applicable

Do you have any ethical concerns with this paper?

No

Have you any concerns about statistical analyses in this paper?

No

Recommendation?

Major revision is needed (please make suggestions in comments)

Comments to the Author(s)

1. English should be improved.
2. The most important quantitative results should be added to the Abstract.

3. The novelty of paper should be highlighted.
4. A list of Abbreviation should be prepared.
5. All equations need appropriate reference.
6. The list of boundary conditions should be prepared in a table.
7. What is chemical setting and heat transfer in numerical solution?
8. Did you observe flameless mode in your experiment?
9. How about numerical settings for NO formation? Which type of NO formation was considered?
10. For NO formation and numerical settings, the following documents are suggested to be considered:
 - "Numerical investigation of biogas flameless combustion." Energy conversion and management 81 (2014): 41-50.
 - "Effects of burner configuration on the characteristics of biogas flameless combustion." Combustion Science and Technology 187.8 (2015): 1240-1262.
11. The items that are mentioned in Conclusion should be discussed in results and discussion section.
12. Conclusion is a summary of the work and the most important results.

Review form: Reviewer 3

Is the manuscript scientifically sound in its present form?

Yes

Are the interpretations and conclusions justified by the results?

Yes

Is the language acceptable?

Yes

Is it clear how to access all supporting data?

Yes

Do you have any ethical concerns with this paper?

No

Have you any concerns about statistical analyses in this paper?

No

Recommendation?

Major revision is needed (please make suggestions in comments)

Comments to the Author(s)

This paper entitled "Experimental and numerical studies on combustion characteristics of N₂-diluted CH₄ and O₂ diffusion combustion in a packed bed" evaluates the combustion characteristics in a porous media experimentally and numerically. The reviewer considers the following comments should be addressed in the revised manuscript:

(1) The schematic figure is too simple to imagine your experimental setup. Please write more in detail including the dimensions of the burners in the schematic figure.

(2) The gas velocity and the mass fraction of the fuel were indicated in the initial part of the section 4.1, but the reviewer thinks that the experimental conditions should be summarized in the section 2. Also, please add the definition of symbols of u_g . In addition, please indicate the gas velocity at the inlet and the fractions of O₂ and N₂ in this section (mass fractions of O₂ and N₂ at the inlet could not find in the manuscript).

(3) P7: Reference of the Rosseland approximation model needs to be added.

(4) P7L40: What is "P_1 model" ? Also, a suitable reference for the P_1 model is required here.

(5) What is the reaction model employed in your numerical simulation. Also the reference for the reaction model is required.

(6) In your experiment, the mass fraction of CH₄ was 0.188 and total excess air ratio was set to 1.88. Why did you select this experimental conditions?

(7) Fig. 4: Since the maximum range of the contours were different for all cases, it was difficult to compare with different conditions. So the reviewer considers that the range of the contour should be the same for all cases, although the maximum temperatures were different each other. In addition, it might be useful to understand if the flame pictures were indicated simultaneously in the figure because the bright part due to the flame may correspond to the high temperature region. In addition, the length scale should be indicated in the figure.

(8) P12L19: You explained that the heat transfer in the packed bed with 2.5 mm pellets is stronger than that of 3.5 mm cases. Please add the reason of this consideration.

(9) Fig. 5: There were some wavy line and line feed symbols in the figure. Please remove them. Please also add the length scale in Fig. 5.

(10) Fig. 6: My understanding is that the meaning of the symbol "h" is the parameter for the height of the pellets, and the maximum value for h was 200 in your study as described in P7L5. However, Fig. 6 shows that the maximum value of examined h value was 240 mm. Please confirm about the description about h.

(11) Fig. 6: According to the reviewer's knowledge, gas sampling for NO may influence of the materials of the sampling probe and immediate reaction quenching at the sampling point is important for accurate measurement [R1]. So please describe the detail structure of the sampling probe and sampling method for NO in your experiment. Also, is it the point measurement or spatially averaged value at the height of h? What is the height of the sampling point from the burner inlet? How is the uniformity of the product gas on the cross section at the height of which gas sampling was conducted?

(12) P13L29: "xxx 2.5 mm is greater than that of d = 2.5 mm when h<120 xxx" may be probably "d = 3.5 mm when xxxx".

(13) P13L35: "packed bed on CO emission is opposite to that of NO_x emission." Since the discussion here is focus on NO, the description of "NO_x" may be changed to "NO".

(14) Fig. 7: The temperature at y = 0 seems to be high (it looks about 1000 K) and the value seems to be increased with an increase in the height of the pallet. The reviewer understand that the packed bed expands the high temperature region. Could you explain why the temperature at the burner inlet is very high even in the case of inlet mixture temperature of 300 K as this study? Also, is it the reasonable temperature compared to the experiment?

Reference

[R1] C. England, J. Houseman, D.P. Teixeira, Sampling nitric oxide from combustion gases, Combustion and Flame, Vol. 20, 1973, 439-442.

Decision letter (RSOS-190492.R0)

23-May-2019

Dear Dr Mao,

The editors assigned to your paper ("Experimental and numerical studies on combustion characteristics of N₂-diluted CH₄ and O₂ diffusion combustion in a packed bed") have now received comments from reviewers. We would like you to revise your paper in accordance with the referee and Associate Editor suggestions which can be found below (not including confidential reports to the Editor). Please note this decision does not guarantee eventual acceptance.

Please submit a copy of your revised paper before 15-Jun-2019. Please note that the revision deadline will expire at 00.00am on this date. If we do not hear from you within this time then it will be assumed that the paper has been withdrawn. In exceptional circumstances, extensions may be possible if agreed with the Editorial Office in advance. We do not allow multiple rounds of revision so we urge you to make every effort to fully address all of the comments at this stage. If deemed necessary by the Editors, your manuscript will be sent back to one or more of the original reviewers for assessment. If the original reviewers are not available, we may invite new reviewers.

- Data accessibility

If you wish to submit your supporting data or code to Dryad (<http://datadryad.org/>), or modify your current submission to dryad, please use the following link:
<http://datadryad.org/submit?journalID=RSOS&manu=RSOS-190492>

- **Competing interests**

- **Authors' contributions**

- **Acknowledgements**

- **Funding statement**

Kind regards,

Andrew Dunn

on behalf of Prof R. Kerry Rowe (Subject Editor)

Associate Editor's comments:

The reviewers have a number of concerns regarding your paper: please ensure that you not only fully respond to these matters but also provide a point-by-point response to them. Also, the language of the paper should be fully revised (<https://royalsociety.org/journals/authors/language-polishing/>) before resubmitting - please ensure you include evidence to show that the language has been addressed.

Comments to Author:

Reviewers' Comments to Author:

Reviewer: 1

Comments to the Author(s)

The manuscript reports experimental and numerical investigations on the diffusion filtration combustion in a pellet packed bed for N₂-diluted CH₄ and O₂. The findings in the paper are interesting. The study started with experimental investigation on flame structure, temperature distribution and NO_x emission. Both immersed flame and surface flame were observed in the experiment. Numerical simulation was conducted to reveal the NO_x formation mechanism with different bed lengths.

The manuscript was well organized and written with fluent English. Publication is recommended after minor revisions. My specific comments are listed below.

1. Please provide a table to include all the experimental parameters and operating conditions.
2. Some grammatical errors are found in the article, such as "It can be seen from the figure that the flame shape of diffusion combustion in porous medium is similar to that of free space, similar to that of conical flame."
3. The authors stated that "In order to consider the thermal conductivity and radiation of quartz glass tube...", however, the energy equation for the quartz glass tube was not included in the mathematics model.

Reviewer: 2

Comments to the Author(s)

1. English should be improved.
2. The most important quantitative results should be added to the Abstract.
3. The novelty of paper should be highlighted.
4. A list of Abbreviation should be prepared.
5. All equations need appropriate reference.
6. The list of boundary conditions should be prepared in a table.
7. What is chemical setting and heat transfer in numerical solution?
8. Did you observe flameless mode in your experiment?
9. How about numerical settings for NO formation? Which type of NO formation was considered?
10. For NO formation and numerical settings, the following documents are suggested to be considered:
 - "Numerical investigation of biogas flameless combustion." Energy conversion and management 81 (2014): 41-50.
 - "Effects of burner configuration on the characteristics of biogas flameless combustion." Combustion Science and Technology 187.8 (2015): 1240-1262.
11. The items that are mentioned in Conclusion should be discussed in results and discussion section.
12. Conclusion is a summary of the work and the most important results.

Reviewer: 3

Comments to the Author(s)

This paper entitled "Experimental and numerical studies on combustion characteristics of N₂-diluted CH₄ and O₂ diffusion combustion in a packed bed" evaluates the combustion

characteristics in a porous media experimentally and numerically. The reviewer considers the following comments should be addressed in the revised manuscript:

- (1) The schematic figure is too simple to imagine your experimental setup. Please write more in detail including the dimensions of the burners in the schematic figure.
- (2) The gas velocity and the mass fraction of the fuel were indicated in the initial part of the section 4.1, but the reviewer thinks that the experimental conditions should be summarized in the section 2. Also, please add the definition of symbols of u_g . In addition, please indicate the gas velocity at the inlet and the fractions of O₂ and N₂ in this section (mass fractions of O₂ and N₂ at the inlet could not find in the manuscript).
- (3) P7: Reference of the Rosseland approximation model needs to be added.
- (4) P7L40: What is "P_1 model" ? Also, a suitable reference for the P_1 model is required here.
- (5) What is the reaction model employed in your numerical simulation. Also the reference for the reaction model is required.
- (6) In your experiment, the mass fraction of CH₄ was 0.188 and total excess air ratio was set to 1.88. Why did you select this experimental conditions?
- (7) Fig. 4: Since the maximum range of the contours were different for all cases, it was difficult to compare with different conditions. So the reviewer considers that the range of the contour should be the same for all cases, although the maximum temperatures were different each other. In addition, it might be useful to understand if the flame pictures were indicated simultaneously in the figure because the bright part due to the flame may correspond to the high temperature region. In addition, the length scale should be indicated in the figure.
- (8) P12L19: You explained that the heat transfer in the packed bed with 2.5 mm pellets is stronger than that of 3.5 mm cases. Please add the reason of this consideration.
- (9) Fig. 5: There were some wavy line and line feed symbols in the figure. Please remove them. Please also add the length scale in Fig. 5.
- (10) Fig. 6: My understanding is that the meaning of the symbol "h" is the parameter for the height of the pellets, and the maximum value for h was 200 in your study as described in P7L5. However, Fig. 6 shows that the maximum value of examined h value was 240 mm. Please confirm about the description about h.
- (11) Fig. 6: According to the reviewer's knowledge, gas sampling for NO may influence of the materials of the sampling probe and immediate reaction quenching at the sampling point is important for accurate measurement [R1]. So please describe the detail structure of the sampling probe and sampling method for NO in your experiment. Also, is it the point measurement or spatially averaged value at the height of h? What is the height of the sampling point from the burner inlet? How is the uniformity of the product gas on the cross section at the height of which gas sampling was conducted?
- (12) P13L29: "xxx 2.5 mm is greater than that of d = 2.5 mm when h<120 xxx" may be probably "d = 3.5 mm when xxxx".
- (13) P13L35: "packed bed on CO emission is opposite to that of NO_x emission." Since the discussion here is focus on NO, the description of "NO_x" may be changed to "NO".
- (14) Fig. 7: The temperature at y = 0 seems to be high (it looks about 1000 K) and the value seems to be increased with an increase in the height of the pallet. The reviewer understand that the packed bed expands the high temperature region. Could you explain why the temperature at the burner inlet is very high even in the case of inlet mixture temperature of 300 K as this study? Also, is it the reasonable temperature compared to the experiment?

Reference

[R1] C. England, J. Houseman, D.P. Teixeira, Sampling nitric oxide from combustion gases, *Combustion and Flame*, Vol. 20, 1973, 439-442.

Author's Response to Decision Letter for (RSOS-190492.R0)

See Appendix A.

RSOS-190492.R1 (Revision)

Review form: Reviewer 2

Is the manuscript scientifically sound in its present form?

No

Are the interpretations and conclusions justified by the results?

Yes

Is the language acceptable?

No

Do you have any ethical concerns with this paper?

No

Have you any concerns about statistical analyses in this paper?

No

Recommendation?

Accept as is

Comments to the Author(s)

Accept

Review form: Reviewer 3

Is the manuscript scientifically sound in its present form?

Yes

Are the interpretations and conclusions justified by the results?

Yes

Is the language acceptable?

Yes

Do you have any ethical concerns with this paper?

No

Have you any concerns about statistical analyses in this paper?

No

Recommendation?

Accept with minor revision (please list in comments)

Comments to the Author(s)

Please confirm following minor comments:

P13L45: Please add proper reference for the equation of h_v . Also, please add the definition of the symbol of " Nu_v ".

P15L20: "that of $d=2.5$." may be " $d=2.5$ mm."

P16L30: " $T_{g,Max}$ " should be " $T_{g,max}$ " (Consistency of "M")

P17L10: You mentioned that "clearly seen in Fig. 6" but the reviewer thinks that the figure may be "Fig. 7". Please confirm this point.

The data supplied as Excel format includes the data for $h = 240$ mm. Please revise and remove the data.

Decision letter (RSOS-190492.R1)

17-Jul-2019

Dear Dr Mao:

On behalf of the Editors, I am pleased to inform you that your Manuscript RSOS-190492.R1 entitled "Experimental and numerical studies on combustion characteristics of N₂-diluted CH₄ and O₂ diffusion combustion in a packed bed" has been accepted for publication in Royal Society Open Science subject to minor revision in accordance with the referee suggestions. Please find the referees' comments at the end of this email.

The reviewers and Subject Editor have recommended publication, but also suggest some minor revisions to your manuscript. Therefore, I invite you to respond to the comments and revise your manuscript.

- Ethics statement

- Data accessibility

It is a condition of publication that all supporting data are made available either as supplementary information or preferably in a suitable permanent repository. The data accessibility section should state where the article's supporting data can be accessed. This section should also include details, where possible of where to access other relevant research materials such as statistical tools, protocols, software etc can be accessed. If the data has been deposited in an external repository this section should list the database, accession number and link to the DOI for all data from the article that has been made publicly available. Data sets that have been

deposited in an external repository and have a DOI should also be appropriately cited in the manuscript and included in the reference list.

If you wish to submit your supporting data or code to Dryad (<http://datadryad.org/>), or modify your current submission to dryad, please use the following link:
<http://datadryad.org/submit?journalID=RSOS&manu=RSOS-190492.R1>

- **Competing interests**

- **Authors' contributions**

- **Acknowledgements**

- **Funding statement**

Because the schedule for publication is very tight, it is a condition of publication that you submit the revised version of your manuscript before 26-Jul-2019. Please note that the revision deadline will expire at 00.00am on this date. If you do not think you will be able to meet this date please let me know immediately.

When submitting your revised manuscript, you will be able to respond to the comments made by the referees and upload a file "Response to Referees" in "Section 6 - File Upload". You can use this

to document any changes you make to the original manuscript. In order to expedite the processing of the revised manuscript, please be as specific as possible in your response to the referees.

Kind regards,

on behalf of R. Kerry Rowe (Subject Editor)
openscience@royalsociety.org

Editor Comments to Author:

The more critical reviewers from the initial submission of your paper have now reported on this revision. They are broadly in favour of publication, but, as you will see, one of the reviewers has a number of remaining modifications they would like you to incorporate before the paper is ready for acceptance. Please ensure you address these concerns in the revised paper, including a point-by-point response, and a tracked-changes version of the paper to help the editors identify the changes you've made.

Reviewer comments to Author:

Reviewer: 3

Please confirm following minor comments:

P13L45: Please add proper reference for the equation of h_v . Also, please add the definition of the symbol of " Nu_v ".

P15L20: "that of $d=2.5$." may be " $d=2.5$ mm."

P16L30: " $T_{g,Max}$ " should be " $T_{g,max}$ " (Consistency of "M")

P17L10: You mentioned that "clearly seen in Fig. 6" but the reviewer thinks that the figure may be "Fig. 7". Please confirm this point.

The data supplied as Excel format includes the data for $h = 240$ mm. Please revise and remove the data.

Author's Response to Decision Letter for (RSOS-190492.R1)

See Appendix B.

Decision letter (RSOS-190492.R2)

30-Jul-2019

Dear Dr Mao,

I am pleased to inform you that your manuscript entitled "Experimental and numerical studies on combustion characteristics of N₂-diluted CH₄ and O₂ diffusion combustion in a packed bed" is now accepted for publication in Royal Society Open Science.

on behalf of Prof R. Kerry Rowe (Subject Editor)
openscience@royalsociety.org

Appendix A

MS Reference Number: RSOS-190492

MS Dryad ID: RSOS-190492

MS Title: Experimental and numerical studies on combustion characteristics of N₂-diluted CH₄ and O₂ diffusion combustion in a packed bed

MS Authors: Li, Houping; Shi, Junrui; Mao, Mingming; Liu, Yongqi

Contact Author: Mingming Mao

Article Type: Original Research Paper

Dear Editor and Reviewers,

We appreciate very much the helpful comments by the reviewers and the editor. The English is also carefully checked and polished. Following is the responses to all comments point by point. We numbered the comments and gave answers. All the revised parts or added content are distinguished by yellow base. Our responses on their questions and suggestions are as follows.

Reviewer: 1

Comments to the Author(s)

The manuscript reports experimental and numerical investigations on the diffusion filtration combustion in a pellet packed bed for N₂-diluted CH₄ and O₂. The findings in the paper are interesting. The study started with experimental investigation on flame structure, temperature

distribution and NO_x emission. Both immersed flame and surface flame were observed in the experiment. Numerical simulation was conducted to reveal the NO_x formation mechanism with different bed lengths.

The manuscript was well organized and written with fluent English. Publication is recommended after minor revisions. My specific comments are listed below.

1. Please provide a table to include all the experimental parameters and operating conditions.

ANSWER: We accept this comment. We added a table (table 3) to list all the experimental parameters and operating conditions.

2. Some grammatical errors are found in the article, such as “It can be seen from the figure that the flame shape of diffusion combustion in porous medium is similar to that of free space, similar to that of conical flame.”

ANSWER: we are sorry for the mistake. We have rewritten the phrase as “It can be seen from the figure that the flame shape of diffusion combustion in porous medium is conical flame, similar to that of free space.”, please see line 57, page 11.

3. The authors stated that “In order to consider the thermal conductivity and radiation of quartz glass tube...”, however, the energy equation for the quartz glass tube was not included in the mathematics model.

ANSWER: We accept this comment. The energy equation for the quartz

glass tube was added in the mathematics model, please see line 58,page 10.

Reviewer: 2

Comments to the Author(s)

1. English should be improved.

ANSWER: We accept this comment. We have done every effort to improve English. According to the editor suggestion, the language of the paper was fully revised via (<https://royalsociety.org/journals/authors/language-polishing/>), please see the evidence as the following picture.

EDITORIAL CERTIFICATE

This document certifies that the manuscript below was edited for correct English language usage, grammar, punctuation and spelling by qualified native English speaking editors at The Charlesworth Group.

Paper Title:

Experimental and numerical studies on combustion characteristics of N₂-diluted CH₄ and O₂ diffusion combustion in a packed bed

Author:

houping li

Date certificate issued:

June 05, 2019

cwauthors.com

2. The most important quantitative results should be added to the Abstract.

ANSWER: We accept this comment. We have added quantitative results in the abstract, please see the revised abstract.

3. The novelty of paper should be highlighted.

ANSWER: We accept this comment. Please see the revised paper in the abstract part.

4. A list of Abbreviation should be prepared.

ANSWER: We accept this comment. We added a new table to list Abbreviation, please see table 1.

5. All equations need appropriate reference.

ANSWER: We accept this comment. The reference for the equations was cited. Please see line 58, page 8.

6. The list of boundary conditions should be prepared in a table.

ANSWER: We accept this comment. Please see Table 2.

7. What is chemical setting and heat transfer in numerical solution?

ANSWER: we added comment on chemical setting and heat transfer in numerical solution. For chemical setting, please see line 50, page 9. "The chemical mechanism for methane combustion provided by Fluent 15.0 is used in this work. To save computation cost, one-step chemistry rather than detailed chemical mechanism is applied, thus simulation of CO and NO_x formation is not considered in this study. The reaction rate is computed as $\omega_{\text{CH}_4} = A(Y_{\text{CH}_4} / W_{\text{CH}_4})^{0.2} (Y_{\text{O}_2} / W_{\text{O}_2})^{1.3} \exp(-E / RT_g)$, A is pre-exponential factor with consistent units and its value is 2.119e11. E is

activation energy ($2.027e8\text{J/Kgmol}$), R is universal gas constant.”

For heat transfer setting, in table 3 we also added thermal properties of porous media and quartz tube, please see table 3.

8. Did you observe flameless mode in your experiment?

ANSWER: No. The flameless mode was not observed in our experiment.

9. How about numerical settings for NO formation? Which type of NO formation was considered?

ANSWER: We presented our preliminary numerical study on diffusion combustion in porous media and one-step chemistry was used in our model, thus simulation of NO was not considered in this work. To clarify this issue, we added comment on the methane combustion mechanism used in this work, please see the answer to question 7.

10. For NO formation and numerical settings, the following documents are suggested to be considered:

- "Numerical investigation of biogas flameless combustion." Energy conversion and management 81 (2014): 41-50.
- "Effects of burner configuration on the characteristics of biogas flameless combustion." Combustion Science and Technology 187.8 (2015): 1240-1262.

ANSWER: The flameless mode was not observed in our experiment. Meanwhile, NO_x formation was not considered in the simulation, thus we did not cite the reference suggested by the reviewer.

10. The items that are mentioned in Conclusion should be discussed in results and discussion section.

ANSWER: We accept this comment. We checked the conclusion part point by point.

11. Conclusion is a summary of the work and the most important results.

ANSWER: We accept this comment. We have rewritten the conclusion, please see line 28, page17.

Reviewer: 3

Comments to the Author(s)

This paper entitled "Experimental and numerical studies on combustion characteristics of N₂-diluted CH₄ and O₂ diffusion combustion in a packed bed" evaluates the combustion characteristics in a porous media experimentally and numerically. The reviewer considers the following comments should be addressed in the revised manuscript:

1. The schematic figure is too simple to imagine your experimental setup.

Please write more in detail including the dimensions of the burners in the schematic figure.

ANSWER: We accept this comment. The Fig. 1 was revised and the dimensions of the burners were added. Please see the revised Fig. 1.

2. The gas velocity and the mass fraction of the fuel were indicated in the initial part of the section 4.1, but the reviewer thinks that the

experimental conditions should be summarized in the section 2. Also, please add the definition of symbols of u_g . In addition, please indicate the gas velocity at the inlet and the fractions of O₂ and N₂ in this section (mass fractions of O₂ and N₂ at the inlet could not find in the manuscript).

ANSWER: We accept this comment. We added a new table 1, 2, 3 to list the symbol, computation cases and boundary conditions, please see table 1-2.

3. P7: Reference of the Rosseland approximation model needs to be added.

ANSWER: We accept this comment. Please see line 40, page 8.

4. P7L40: What is "P₁ model" ? Also, a suitable reference for the P₁ model is required here.

ANSWER: We are sorry for the mistake. Here "P₁ model" should be "Discrete model", please see line 45, page 8.

5. What is the reaction model employed in your numerical simulation.

Also the reference for the reaction model is required.

ANSWER: This comment is same to the question 7 by Reviewer 2. Please see the answer to this issue above.

6. In your experiment, the mass fraction of CH₄ was 0.188 and total excess air ratio was set to 1.88. Why did you select this experimental conditions?

ANSWER: We prefer to study fuel-lean diffusion combustion in porous media. At the same time, a smaller mixture velocity is selected to prevent destroying the quartz tube, as we can see that the combustion temperature increases as the mixture velocity is increased in the experiment. After many experiments, we select the mentioned experimental conditions.

7. Fig. 4: Since the maximum range of the contours were different for all cases, it was difficult to compare with different conditions. So the reviewer considers that the range of the contour should be the same for all cases, although the maximum temperatures were different each other. In addition, it might be useful to understand if the flame pictures were indicated simultaneously in the figure because the bright part due to the flame may correspond to the high temperature region. In addition, the length scale should be indicated in the figure.

ANSWER: We accept this comment. The color scale for the temperature is set to be same for all the figures. However, the length scale for the high temperature zone is very difficult to define, it is a qualitative figure in length direction.

8. P12L19: You explained that the heat transfer in the packed bed with 2.5 mm pellets is stronger than that of 3.5 mm cases. Please add the reason of this consideration.

ANSWER: We accept this comment. We add comment on this issue, please see line 43, page 13.

9. Fig. 5: There were some wavy line and line feed symbols in the figure.

Please remove them. Please also add the length scale in Fig. 5.

ANSWER: We accept this comment. Please see the revised Fig. 5.

10. Fig. 6: My understanding is that the meaning of the symbol "h" is the parameter for the height of the pellets, and the maximum value for h was 200 in your study as described in P7L5. However, Fig. 6 shows that the maximum value of examined h value was 240 mm. Please confirm about the description about h.

ANSWER: We are sorry for the mistake. In the revised manuscript "h" was defined in table 1. It was right that the meaning of the symbol "h" was the parameter for the height of the pellets, and the maximum value for h was 200 in our study, as we have stated in the original paper. However, we have done the experimental studies for $h=240$ mm for some experiment cases and we made mistake to show it in the figure. In the revised paper, the experimental data for $h=240$ mm was deleted. Please see the revised Fig. 6.

11. Fig. 6: According to the reviewer's knowledge, gas sampling for NO may influence of the materials of the sampling probe and immediate reaction quenching at the sampling point is important for accurate measurement [R1]. So please describe the detail structure of the sampling probe and sampling method for NO in your experiment. Also, is it the point measurement or spatially averaged value at the height of

h? What is the height of the sampling point from the burner inlet?
How is the uniformity of the product gas on the cross section at the height of which gas sampling was conducted?

ANSWER: This is a good suggestion. In the original manuscript we didn't state clear the sampling method and position. We added comment on this issue. "Exhaust gas is collected using a stainless steel probe placed at the top of the combustor, the concentrations of NO_x and CO in the exhaust from the combustor are measured. The height of the combustor and the maximum porous media length is 490 mm and 200 mm, respectively. It is observed in the experiment that the maximum flame height is about 200 mm for all investigated cases, this means that the exhaust gas is well premixed and the chemical reactions come to a halt before entering the stainless steel. Thus the measured NO_x and CO is spatially averaged values.", please see line 23, page 14.

12.P13L29: "xxx 2.5 mm is greater than that of $d = 2.5$ mm when $h < 120$ xxx" may be probably " $d = 3.5$ mm when xxxx".

ANSWER: We are sorry for the mistake. We have corrected this issue, please see line 14, page 15.

13. P13L35: "packed bed on CO emission is opposite to that of NO_x emission." Since the discussion here is focus on NO, the description of "NO_x" may be changed to "NO".

ANSWER: We are sorry for the mistake. "NO_x" was replaced by "NO".

Please see line 19, page 15.

14. Fig. 7: The temperature at $y = 0$ seems to be high (it looks about 1000 K) and the value seems to be increased with an increase in the height of the pallet. The reviewer understand that the packed bed expands the high temperature region. Could you explain why the temperature at the burner inlet is very high even in the case of inlet mixture temperature of 300 K as this study? Also, is it the reasonable temperature compared to the experiment?

ANSWER: We accept this comment. We added comment on this issue, please see line 12, page 16. In addition, the predictions were not validated against the experimental data. We measured the external walls temperature of quartz tube by the infrared thermal imager. The temperature within the packed bed was not measured in the experiment, thus it was difficult to perform comparison between the experiments and predictions. We presented our preliminary study and hoped the reviewer understanding.

Reference

[R1] C. England, J. Houseman, D.P. Teixeira, Sampling nitric oxide from combustion gases, *Combustion and Flame*, Vol. 20, 1973, 439-442.

Appendix B

MS Reference Number: RSOS-190492

MS Dryad ID: RSOS-190492

MS Title: Experimental and numerical studies on combustion characteristics of N₂-diluted CH₄ and O₂ diffusion combustion in a packed bed

MS Authors: Li, Houping; Shi, Junrui; Mao, Mingming; Liu, Yongqi

Contact Author: Mingming Mao

Article Type: Original Research Paper

Dear Editor and Reviewers,

We appreciate very much the helpful comments by the reviewers and the editor. The English is also carefully checked and polished. Following is the responses to all comments point by point. We numbered the comments and gave answers. All the revised parts or added content are distinguished by yellow base. Our responses on their questions and suggestions are as follows.

Reviewer: 3

1. Please confirm following minor comments: P13L45: Please add proper reference for the equation of h_v . Also, please add the definition of the symbol of " Nu_v ".

ANSWER: We accept the comment. We are sorry for the mistake. The proper reference for the equation of h_v has been added, please see 43

line , 13 page . The definition of the symbol of "Nu_v" has been presented, please see 48 line , 13 page .

2. P15L20: "that of d=2.5." may be "d=2.5 mm."

ANSWER: We accept the comment. We are sorry for the mistake. The "d=2.5" has been replaced by "d=2.5 mm", Please see 24 line , 15 page .

3. P16L30: "T_{g,Max}" should be "T_{g,max}" (Consistency of "M")

ANSWER: The "T_{g,Max}" has been replaced by "T_{g,max}". Please see 33 line , 16 page .

4.P17L10: You mentioned that "clearly seen in Fig. 6" but the reviewer thinks that the figure may be "Fig. 7". Please confirm this point.

ANSWER: Yes, you are right. We are sorry for the mistake. The "Fig.6" has been replaced by "Fig. 7". Please see 11 line , 17 page .

5.The data supplied as Excel format includes the data for $h = 240$ mm. Please revise and remove the data.

ANSWER: We accept the comment. We are sorry for the mistake. The data for $h=240$ mm has been removed. Please see data accessibility part.